# µeV electron spectromicroscopy using free-space light

Yves Auad [1], Eduardo J. C. Dias [2], Marcel Tencé[1], Jean-Denis Blazit[1], Xiaoyan Li[1], Luiz Fernando Zagonel [3], Odile Stéphan[1], Luiz H. G. Tizei [1], F. Javier García de Abajo [2,4] ✉ & Mathieu Kociak [1] ✉

The synergy between free electrons and light has recently been leveraged to reach an impressive degree of simultaneous spatial and spectral resolution, enabling applications in microscopy and quantum optics. However, the required combination of electron optics and light injection into the spectrally narrow modes of arbitrary specimens remains a challenge. Here, we demonstrate microelectronvolt spectral resolution with a sub-nanometer probe of photonic modes with quality factors as high as $10^4$. We rely on mode matching of a tightly focused laser beam to whispering gallery modes to achieve a $10^8$-fold increase in light-electron coupling efficiency. By adapting the shape and size of free-space optical beams to address specific physical questions, our approach allows us to interrogate any type of photonic structure with unprecedented spectral and spatial detail.

Thanks to a sustained series of impressive advances in instrumentation[1–5], electron microscopes can currently focus 60–300 keV electrons down to sub-Ångström focal spots with an energy spread of just a few meV. As impressive as this spectral resolution might seem, optical modes of high quality factor $Q$, which are of utmost importance for applications including quantum optics and optical metrology, possess substantially smaller linewidths and, therefore, are unresolvable by state-of-the-art electron spectroscopies such as energy-loss spectroscopy (EELS), cathodoluminescence (CL)[6–8], and photon-induced near-field electron microscopy (PINEM)[9–13].

The so-called electron energy-gain spectroscopy (EEGS) was proposed[14] as a technique that can dramatically enhance electron-based spectroscopies by inheriting the spectral resolution of laser sources while retaining the spatial resolution of electron beams (e-beams). EEGS data thus consist of a series of conventional EELS spectra that are acquired as one scans the wavelength of an external laser irradiating the specimen. Electron-light coupling is mediated by near-field optical components, whose strength is dependent on the optical response of the sample. The latter is consequently retrieved from the intensity associated with light-induced electron energy-gain events as a function of laser wavelength[14,15], with an energy resolution that is only limited by the energy-time photon uncertainty (~$\hbar$ ~ 1 eV fs). Early attempts to demonstrate EEGS measurements were performed with 100s-fs laser pulses[16,17], therefore reaching a spectral resolution of tens of meV, yet not overtaking the few-meV resolution of modern electron monochromators[2]. In parallel, efforts were undertaken to develop EEGS using nanosecond[18] or continuous-wave[19] visible laser sources combined with continuous e-beams, although these attempts did not introduce any wavelength tunability. Recently, a spectral resolution of a few µeV has been demonstrated using on-purpose designed photonic waveguides operating in the near-infrared[20]. By injecting continuous laser light into the near field of the structure, this work elegantly circumvented the fundamental problem of optical coupling to a high-$Q$ cavity in the far field. Indeed, the fact that $Q$ is high essentially boils down to the lack of efficient radiative coupling to the far field. Nevertheless, a sample holder equipped with a dedicated optical fiber was needed to achieve efficient coupling to a photonic device of 10s µm in size, thus limiting its applicability to a limited range of specimens.

Here, we demonstrate high-spectral-resolution EEGS using free-space light injection enhanced by mode matching between a

[1]Université Paris-Saclay, CNRS, Laboratoire de Physique des Solides, 91405 Orsay, France. [2]ICFO-Institut de Ciencies Fotoniques, The Barcelona Institute of Science and Technology, 08860 Castelldefels, Barcelona, Spain. [3]Gleb Wataghin Institute of Physics, University of Campinas - UNICAMP, 13083-859 Campinas, SP, Brazil. [4]ICREA-Institució Catalana de Recerca i Estudis Avançats, Passeig Lluís Companys 23, 08010 Barcelona, Spain. ✉e-mail: javier.garciadeabajo@nanophotonics.es; mathieu.kociak@universite-paris-saclay.fr

free-space laser beam and the sample, which renders the technique applicable to a broad variety of photonic specimens. Specifically, we excite whispering-gallery modes (WGMs) of well-defined angular momenta in spherical resonators by means of a focused off-axis laser Gaussian beam using a high-numerical-aperture mirror. We first concentrate on ~4 μm silica spheres with quality factors $Q \sim 100$–300, in which EELS and CL characterization reveals sharp resonances[21] that are corroborated by EEGS with a higher spectral resolution of ~2 meV. This test system shows that optimum laser-mode coupling is achieved via the conservation of angular momentum, resulting in a $10^8$-fold enhancement of the coupling efficiency relative to irradiation by an unfocused light plane wave. The improved sensitivity of EEGS is also explained in terms of the sharp laser linewidth (7 μeV) and high numerical aperture of the light injection system. We then demonstrate the full potential of this approach by controlling the laser beam position with sub-μm accuracy and resolving narrow optical modes ($Q \sim 10^4$) in EEGS spectra of ~8 μm polystyrene spheres, which are unobservable by EELS or CL. Our technique is readily applicable to study arbitrary photonic structures and represents an increase by more than two orders of magnitude in spectral resolution relative to state-of-the-art EELS with the same spatial resolution.

## Results

### Experimental setup
A sketch of the experimental setup is displayed in Fig. 1. Experiments were carried out in a modified Nion Hermes 200 transmission electron microscope (Chromatem) working at 200 keV with a sub-nanometer e-

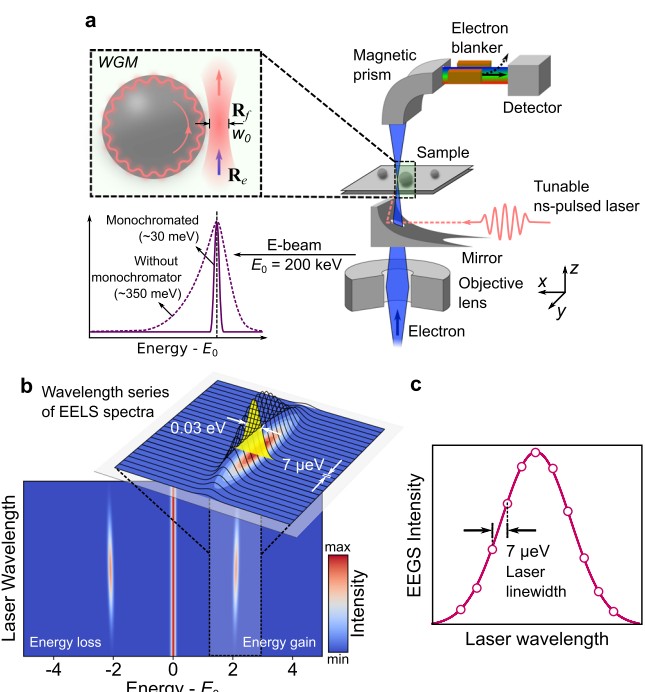

**Fig. 1 | Versatile EEGS experiments using a continuous electron gun microscope. a** Schematics of the setup. A laser beam is focused down to a ~1 μm spot and positioned with sub-μm precision at $\mathbf{R}_f$ on the sample of interest by using a parabolic mirror. A monochromatized 200 keV electron is also focused on the sample and scanned to acquire EELS, EEGS, and CL signals. EEGS measurements are taken by synchronizing light (using a laser trigger) and electrons (through an e-beam blanker) at the detector. **b** A series of EELS spectra is acquired for a given electron probe position as the laser wavelength is scanned over the spectral region of interest (simulated data). The strength of the energy-gain signal is boosted when the light wavelength is close to an optical resonance of the specimen. **c** Schematic of a resonance probed by EEGS in a constant laser power regime. The energy-gain signal yields the EEGS intensity as a function of light wavelength, only limited by the nominal laser linewidth (7 μeV for the laser used in this study).

beam. The choice of this high electron acceleration voltage was dictated by the need for phase matching between the WGM and the electron[21], not to be confused with mode matching between light and the WGM, which is discussed below. A nanosecond-laser beam was focused on the sample and synchronized with the EELS detection in order to record PINEM spectra, as detailed in the Supplementary Information (SI). Critical improvements were implemented relative to our previous work[18]. Specifically, the microscope was fitted with a high-numerical-aperture (NA ~ 0.5) Attolight Mönch light detection/injection system able to focus down to a 1 μm spot size with sub-μm accuracy (see details of the spot profile in the SI). The time-averaged laser input power of 1–5 mW used in experiments resulted in a typical $10^{8-9}$ W/m² optical spot intensity. We used the mirror to position the laser spot at the edge of the WGM resonators. In addition, the laser wavelength was tuned to spectrally map the resonances, with a wavelength resolution of 2 pm (~7 μeV at 585 nm), limited by the laser specifications. Because the EEGS signal was weak (~$10^{-4}$ of the measured ZLP), a large increase of signal-to-noise ratio was needed, which we achieved through a slight monochromation of 30–50 meV over the 350 meV initial e-beam energy spread. This led to a strong suppression of the ZLP tails, which would otherwise produce a substantial background. We recorded spectra with a direct electron detector (MerlinEM, from Quantum Detectors) using an effective current of ~0.2 fA (see SI), comparable to PINEM experiments. The WGM resonators were drop-cast on a lacey-carbon sample grid, which was coated with 60 nm of silver to improve charge and thermal dissipation. Finally, alignment of the laser spot to the microscope optical axis was achieved with ~1 μm precision by maximizing the electron EEGS signal from a featureless silver film. A more detailed description of the setup is offered in the SI.

The EEGS electron-light coupling is described by a single parameter[22]:

$$\beta(\mathbf{R}_e,\omega) = \frac{e}{\hbar\omega} \int_{-\infty}^{\infty} dz\, E_z(\mathbf{R}_e,z,\omega)\, e^{-i\omega z/v}, \qquad (1)$$

where $\mathbf{R}_e$ is the transverse electron probe position, $v$ is the electron velocity, $\omega$ is the angular frequency of the external light, and $E_z$ is the optical electric field component along the e-beam direction $z$, corresponding to a time-varying field $E_z(\mathbf{R}_e,t) = 2\mathrm{Re}\{E_z(\mathbf{R}_e,z,\omega)e^{-i\omega t}\}$, which is dependent on the focal beam profile and position relative to the specimen (see SI). In our experiment, we use a low peak-intensity illumination, such that $|\beta|^2 \ll 1$ is the probability for the electron to gain one photon quantum (i.e., the EEGS signal is essentially a perturbation).

### Medium spectral resolution EEGS using 4 μm silica spheres
To illustrate and validate the principle of EEGS and its relation to other spectroscopies, we start by studying ~4 μm silica spheres (Fig. 2), which are known to exhibit under similar experimental conditions quality factors $Q \sim 10^2$ and good electron-WGM coupling for the employed 200 keV electrons[21], and have been successfully studied by PINEM[13]. In Fig. 2a, we plot a measured series of spectra acquired for varying light wavelengths (vertical axis, in steps of 250 pm, corresponding to a 0.92 meV photon-energy interval at 580 nm) with a constant laser power of ~1.5 mW and the e-beam probe positioned as indicated by the blue dot $\mathbf{R}_e$ in Fig. 2c. We observe two distinct WGMs with $Q = 244$ and $Q = 194$ separated by a spectral distance of 66.4 meV. Due to the high monochromaticity of the e-beam, it is possible to resolve the energy-gain resonance shifting in energy as we raster the laser wavelength (dashed white line in Fig. 2a). One of every four of these spectra is shown in the cascade in Fig. 2b, where we note the presence of both stimulated electron energy gain and loss features. The gain signal accounts for a fraction ~$5 \times 10^{-4}$ of the integrated measured spectrum (see color bar in Fig. 2a), implying a

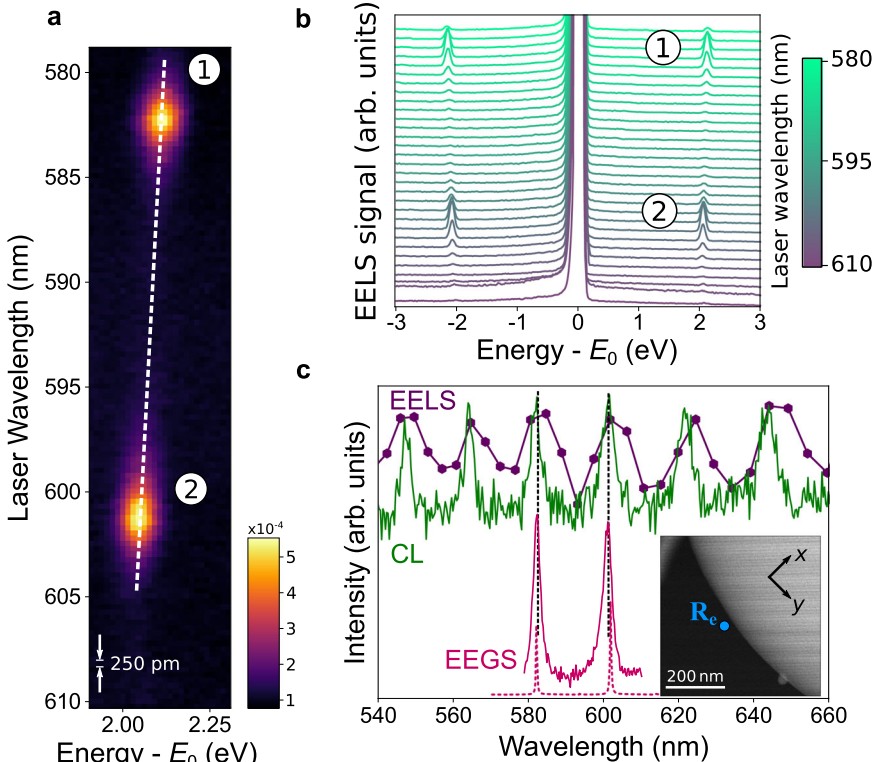

**Fig. 2 | EEGS measurements in WGM resonators with $Q \sim 100$–$300$. a** Measured series of EELS gain spectra as a function of light wavelength (vertical axis), showing two distinct resonances at 586.27 nm (2.1148 eV) and 601.02 nm (2.0629 eV) with quality factors of 244 and 194, respectively. **b** Cascade of one in every four spectra taken from (**a**), where both the stimulated loss and gain sidebands are observable. **c** EEGS, EELS, and CL measured spectra (solid curves) compared to theory (dashed curves) for the same e-beam probe position (blue circle at $\mathbf{R}_e$ in the inset) and mirror focal position.

gain probability of ~$5 \times 10^{-3}$ after correcting for the e-beam blanker time window (~10 times the laser pulse duration). Furthermore, there are no visible higher-order replicas at multiples of the photon energy $\pm n\hbar\omega$, as expected in the low-intensity regime[10,11]. In Fig. 2c, we display the measured EEGS intensity (solid red curve) obtained by integrating the EELS spectra series between 1.9 eV and 2.3 eV for each light wavelength. A similar result is obtained by integrating the stimulated loss peak, although the gain side is free from loss features and thus has a better signal-to-background ratio. In this particular sample, the ~30 meV spectral resolution of our setup is already enough to unveil the same gallery modes in EELS, and additionally, modes can be identified by collecting light leakage from the resonator to the far field to record a CL spectrum. The three spectroscopies (EEGS, EELS, and CL) are performed for the same e-beam probe position $\mathbf{R}_e$ and mirror focal spot $\mathbf{R}_f$. As expected from the reciprocity of Maxwell's equations, the CL and EEGS spectral variations look similar[22] (see SI). Also, in non-dissipative systems, the EELS and CL probabilities should be identical because radiation losses are the only source of energy losses[23], although deviations between the two of them can arise because we are collecting only a fraction of the emission solid angles[24]. Nevertheless, no spectral shifts can be discerned between EELS (the equivalent of optical extinction[8]), CL (scattering), and EEGS within the single-pixel uncertainty of the EELS channels. Theoretical modeling of the EEGS intensity (Fig. 2c, dashed curves; see details in SI) matches the WGM positions for a fitted sphere diameter of 4122 nm (consistent with the experimentally determined diameter), although the predicted quality factors are ~1500, almost one order of magnitude higher than the experimental results, presumably because of the effect of losses produced at the supporting carbon structure. The reported EEGS features with a full width at half maximum (FWHM) of ~2 meV separated by 66.4 meV already demonstrate a spectral resolution one order of magnitude

better than the used EELS resolution at 200 keV electron energy, but also better than the ultimate spectral resolution of the machine (~5 meV at 60 keV).

It should be noted that CL yields a faint signal, which we accumulate for ~30 s without electron monochromation (i.e., using 10–20 times more current than in the monochromated experiments). Although this is faster than EEGS acquisition, the high electron currents used can produce larger sample damage. In addition, the EEGS signal can be enhanced by increasing the incident laser power, which is still well below the sample damage threshold.

## Mode matching

We next interrogate the potential of EEGS for the investigation of high-$Q$ photonic modes (i.e., those in which a high spectral resolution is required). Identifying these modes in a free-space configuration is experimentally challenging, as they are, by definition, weakly coupled to free-space light. Therefore, a clear understanding of how to maximize light coupling is needed. We start by presenting theoretical calculations that illustrate the benefits of using a position-controlled high-numerical-aperture focusing system.

The EEGS simulations presented in Fig. 3a for the $SiO_2$ sphere studied in Fig. 2 show an enhancement in probability by eight orders of magnitude when switching from plane wave illumination to focused illumination (assuming the same power and initial laser beam extension over the mirror area in both scenarios). Besides a clear improvement due to the focusing effect of the mirror, the principle behind such a huge increase in the coupling efficiency can be found in mode-matching between a free-space optical beam and WGMs when the former is focused near the edge of a dielectric sphere[25]. Such a mode-matching can be well understood in terms of a preferential orbital angular number $l \sim 2\pi R_f / \lambda$ produced when the focal spot is at a distance $R_f$ from the sphere center and $\lambda$ is the light wavelength. Near the edge

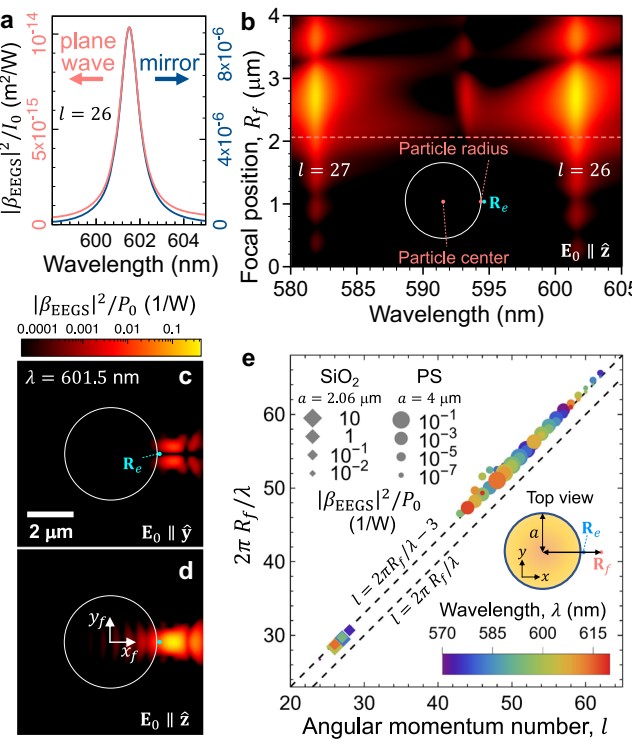

**Fig. 3 | Free-space optical mode matching.** Dependence of the EEGS probability on the illumination conditions for a SiO$_2$ sphere of 4122 nm diameter. **a** EEGS probability around a 601.5 nm resonance for plane-wave and mirror-focused illumination, normalized to the light intensity incident on the sphere and mirror, respectively. **b** Dependence of the EEGS probability on light wavelength and focal spot position. The vertical axis corresponds to the distance from the focal spot to the sphere center as it moves along the $x$ (horizontal) axis (see inset). **c, d** EEGS probability maps at the peak wavelength of (**a**) with $\mathbf{R}_f = (x_f, y_f)$ scanned for incident light polarized along $y$ and $z$, respectively. The sphere contour is shown as a white circle. The color scale is shared by panels (**b**) to (**d**). **e** Optimum optical-focus position $R_f$ as a function of orbital momentum number $l$ for resonances in PS (circles) and SiO$_2$ (diamonds) spheres of diameters 8000 and 4122 nm, respectively. Symbols show all modes of high quality factor within the $\lambda = 570–620$ nm spectral region. The EEGS probability is indicated by the symbol size (see legend). The dashed straight lines correspond to $2\pi R_f/\lambda$ equal to $l$ and $l + 3$. In (**a**), (**b**), and (**e**), the e-beam passes at a fixed position ~80 nm away from the sphere surface on the $x$ axis, as indicated in the insets. The intensity $I_0$ and power $P_0$ of the light incident on the mirror are related through $P_0 = A\,I_0$, where $A = 18.74$ mm$^2$ is the mirror area.

of the silica sphere discussed in Fig. 2, we have $R_f = 2060 \pm 500$ nm, leading to preferential coupling to modes of angular momentum number $l = 22 \pm 5$, in agreement with the angular orders $l = 26$ and 27 deduced from theory for the modes under investigation (see SI). The beneficial effects of mode-matching can be further evidenced in several ways. The EEGS signal for an electron passing near the edge of a dielectric sphere is represented in Fig. 3b as a function of the optical beam position for a fixed wavelength, revealing that the signal is strikingly peaked at the position of optimum mode-matching for the value of $l$ corresponding to the mode that is resonant at a wavelength $\lambda$. In Fig. 3c, d, the mirror position is scanned with a fixed electron probe position $\mathbf{R}_e$ for the two possible light polarizations, again evidencing the mode-matching condition through an optimal mirror position $\mathbf{R}_f$. These results are further corroborated by examining multiple resonances comprised in the 570–620 nm range both for the already discussed 4.122 μm SiO$_2$ sphere and for a bigger 8 μm polystyrene (PS) sphere. The relation between the focal point position and the angular momentum number is linear as expected[25], although shifted by 3, presumably as a result of the finite beam size. More generally, mode

matching between focused light and WGM resonators is known to be similar to light coupling to a waveguide[26] and reach coupling efficiencies up to 20%. We thus anticipate that high-$Q$ cavities could be studied with nanometer-scale resolution in a way similar to waveguides[20], but with the much higher flexibility regarding the type of specimen and the light coupling arrangement enabled by free-space coupling.

### High spectral resolution EEGS using 8 μm polystyrene spheres

To validate this hypothesis, we examined a larger PS sphere of 8 μm in diameter in search of high-$Q$ resonances. In the one considered in Fig. 4, EEGS measurements with a laser power of ~1 mW revealed quality factors as high as $10^4$ (FWHM of 194 μeV). In the wavelength series presented on the left of Fig. 4a, the laser step was fixed at 50 pm, scanning a relatively broad energy range ~88.34 meV from 580 to 605 nm. After identifying a sharp resonance close to 592.6 nm, the energy range and the laser step were reduced to the limit of our laser, rastering a wavelength range of 0.7 nm in steps of 2 pm (i.e., a spectral step of approximately 7 μeV). The yellow dashed rectangle illustrates the energy range used for the sequential acquisitions. This series of measurements demonstrates the potential of EEGS to map a significant range of resonance linewidths by adapting the laser wavelength scan range and the spectral step. Interestingly, the acquisition of each wavelength series took ~8 min, showing remarkable repeatability of the experiments, with minor changes between the displayed series. The wavelength series at the right of Fig. 4a is integrated along the electron energy axis for each laser wavelength to produce the EEGS intensity curve shown in Fig. 4b, as well as the Lorentzian fittings for the three sharpest resonances, yielding quality factors of 7430, 10541, and 9178. The peak separation is ~300 μeV, and the highest-$Q$ mode has a FWHM of 194 μeV.

### Specific advantage of EEGS with respect to EELS and CL

We remark that EELS cannot resolve such fine features due to its limited spectral resolution. Also, although CL could in principle be performed with a sufficiently accurate light spectrometer, it cannot resolve high-$Q$ features in practice because of its smaller signal count rate. Indeed, the ratio of integrated CL and EEGS probabilities scales as $\Gamma_{CL}/\Gamma_{EEGS} \sim 1/Q$ when measuring a mode of quality factor $Q$ (see detailed derivation in the SI). In this work, the illumination intensity is ~$10^8$ W/m$^2$ and the resonance energy ~2 eV, and hence, we have $\Gamma_{CL}/\Gamma_{EEGS} = 2.5$ for $Q = 200$, thus explaining why CL can resolve the WGMs in the smaller silica spheres discussed in Fig. 2. In contrast, $\Gamma_{CL}/\Gamma_{EEGS} = 0.05$ for $Q = 10^4$, therefore yielding an undetectable CL signal in the larger spheres. This constitutes a compelling argument supporting the superior signal-to-noise ratio of EEGS relative to CL. It is worth mentioning that the same arguments limit the sensitivity of EEGS when it is performed with a laser of small spectral resolution compared to the WGM linewidth (e.g., when using femtosecond light pulses, in which most of the injected photons lie outside the resonance, thus resulting in a dramatic loss of coupling efficiency).

### Discussion

The present work demonstrates EEGS with nanometer spatial resolution and down to 200 μeV spectral resolution on arbitrary optical dielectric cavities, therefore leveraging the spatial resolution of free electrons, the versatility of electron microscopy, and the spectral resolution of laser light sources. Our results are made possible by using a small laser bandwidth compared with the widths of the probed modes, as well as by adapting the symmetry, size, and shape of the laser beam to that of the excitations in those cavities, all in a free-space configuration. More general strategies for laser-to-cavity mode matching could rely on light beams sculpted in amplitude and phase through slide projection (e.g., through spatial light modulators). This versatility holds potential for imaging at the ultimate limits of

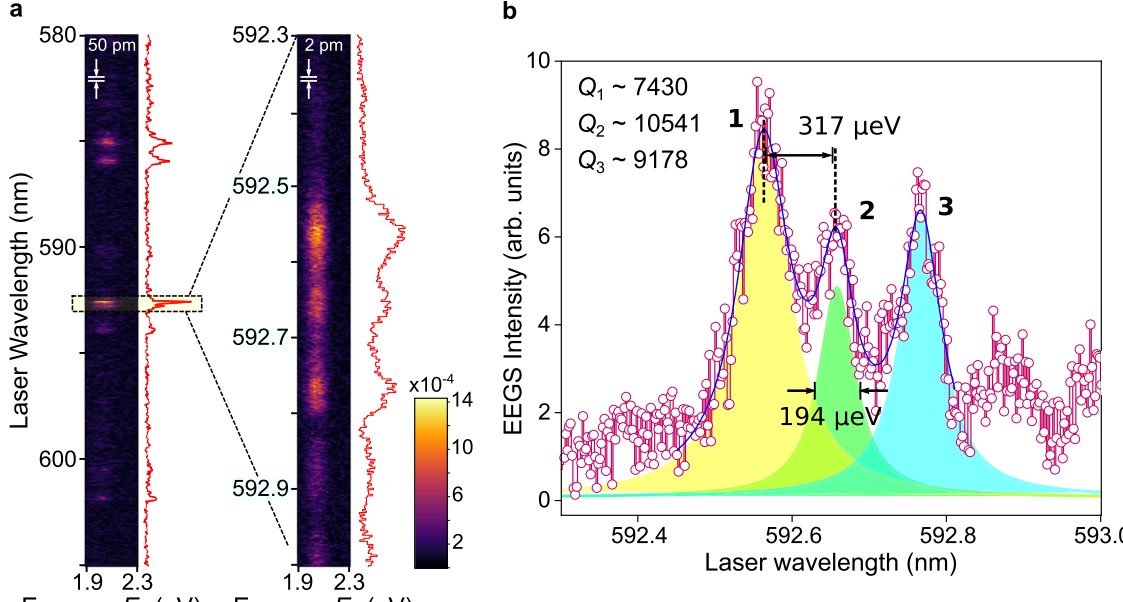

**Fig. 4 | High quality factor in large polystyrene spheres. a** Two laser wavelength series of EELS spectra, taken at the same electron probe position over an increasingly small wavelength range with step sizes of 50 pm and 2 pm, respectively. **b** EEGS spectrum constructed from the rightmost EELS series in (**a**), along with Lorentzian fittings revealing three resonances (1–3) of quality factors 7430, 10541, and 9178, respectively. The uncertainties for the measured quality factor are <1 for the three resonances.

resolution, as required for the characterization of vanguard photonic structures. Looking forward, we further envision the study of optical materials such as quantum-confined systems, 2D crystals, and point defects. Nevertheless, we note that, even in PINEM, no experiments have been reported on optical excitations in atomic-scale systems, presumably because of the low electric field in Eq. (1) expected for excitons or electron-hole pairs compared to collective excitations such as plasmons or other macroscopic photonic modes. Extreme mode- and energy-matching such as we propose here could be the key to the success of such investigations. It should be noted that EEGS is not limited to the visible spectrum, but it should be useful to probe mid-[27] and far-infrared modes, going well beyond the spectral resolution reached by electron monochromation, which is now reaching the limits prophesied by their creators[28]. Applications in vibration mapping at such high resolution should directly impact biological applications[29]. Fast beam blanking technologies are also becoming increasingly available in distinct operation frequencies and duty cycles[30,31], enabling this experiment to be implemented in different microscope configurations. In addition, beyond the current design, energy-gain experiments using pulsed laser sources could be performed without e-beam blankers by relying on time-resolved electron detectors[32]. The use of nanosecond-based time-resolved detectors could further increase the spectroscopic portfolio in electron microscopy by means of temporal correlation between photons and electrons, as recently demonstrated in experiment[33,34]. The integration of such new techniques holds promise for a thriving future in free-electron-based nano-optics.

## Data availability

The data generated in this study have been deposited on zenodo under accession code 7795694.

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

## Acknowledgements

We thank T. Lovejoy for discussions on the functioning of the fast blanker on the Chromatem machine. We are also thankful to H. Lourenço-Martins, S. Meuret, and A. Arbouet for fruitful discussions. This project has been funded in part by the European Union through the Horizon 2020 Research and Innovation Program (grant agreement No. 101017720 (EBEAM)) (M.K., Y.A., O.S., M.T., L.H.G.T., X.L., J.-D.B., and F.J.G.d.A.), the French National Agency for Research under the program of future investment TEMPOS-CHROMATEM (reference No. ANR-10-EQPX-50) (M.K., Y.A., O.S., M.T., L.H.G.T., X.L., and J.-D.B.) and QUENOT (ANR-20-CE30-0033) (M.K., Y.A., O.S., M.T., L.H.G.T., X.L., and J.-D.B.), the Spanish MICINN (PID2020-112625GB-I00 and SEV2015-0522) (F.J.G.d.A. and E.J.C.D.), the ERC (Advanced Grants 789104-eNANO and 787510-4DBIOSERS) (F.J.G.d.A. and E.J.C.D.), the Catalan CERCA Program, and Fundació Privada Cellex (F.J.G.d.A. and E.J.C.D.).

## Author contributions

M.K., Y.A., O.S., M.T., L.H.G.T., and F.J.G.d.A. designed the experiment. Y.A., M.T., and J.-D.B. developed the light injection and synchronization setup with support from L.F.Z. and L.H.G.T. Y.A. performed experiments with support from L.H.G.T., X.L., and M.K. Y.A. analyzed the data with support from L.H.G.T. F.J.G.d.A. developed the theory and carried out numerical simulations with support from E.J.C.D. All authors participated in the results analysis and contributed to the preparation of the manuscript.

## Competing interests

M.K. is a consultant for Attolight. M.K., L.F.Z., M.T., and J.L.D. have licensed know-how and patents to Attolight. M.T. has licensed know-how to Quantum Detectors. Other authors declare no conflicts of interest.
