## [Peer review file · Nature Communications]

REVIEWER COMMENTS

Reviewer #1 (Remarks to the Author):

The manuscript by Auad et al. reports fascinating experiments on electron-energy gain spectroscopy (EEGS) at a resonant dielectric nanostructure. In the experiments, a monochromated electron beam from a designated scanning transmission electron microscope (STEM) is passed by optically illuminated dielectric spheres. These structures facilitate the inelastic interaction of the electrons with excited whispering gallery modes, which results in stimulated scattering and the generation of single sidebands in the electron energy spectrum (as in so-called PINEM experiments).

The present study varies the frequency of the incident nanosecond laser across different resonances in the structure and records the fraction of electrons inelastically scattered, which promotes a higher resolution than that of the microscope by a form of "excitation spectroscopy", which was previously coined EEGS (by two of the authors, in ref. [14]).

The technical development, measured data, and overall manuscript presentation are of excellent quality and, in my opinion, clearly merit publication in Nature Communications. A particularly interesting physical aspect concerns the mode-matched excitation of the dielectric resonators by focusing the laser in close vicinity of the surface. I believe the work will be of considerable interest to readers from both the immediate community and further subject areas.

I suggest the authors address the following minor comments in a revision:

1) The authors state that the approach allows them to "interrogate any type of photonic structure", and that it is "generally applicable to any kind of specimen" and "arbitrary structures". I fully agree with the authors on the impressive progress documented by this work, but these specific phrases might be slight overstatements. I believe it is not a coincidence that the present work considers structures that resemble characteristic features of Refs. [13] and [20]. Specifically, the combination of phase matching and resonant enhancement does seem to be practical or possibly necessary for the observability of the phenomenon with present technical capabilities. Possibly, the authors could slightly balance these formulations and also link their results a bit better to these recent observations.

2) Panel 1c shows data points separated by 7 μeV . It is not fully clear from the figure and caption that this panel is also a schematic and not a measurement, and that the 7 μeV is the linewidth of the laser.

3) The abstract states that the authors carried out a "nanoscale mapping of photonic modes", but the manuscript does not show any spatial scans of the electron beam. Have the authors also carried out such measurements? This should probably be addressed. Can they comment on opportunities and challenges towards EEGS-imaging?

Reviewer #2 (Remarks to the Author):

Manuscript by Y. Auad et al, "microeV electron"

This manuscript presents a novel implementation of the energy gain spectroscopy approach, improving the energy resolution and signals by order of magnitudes compared to

previous work. The work is exciting as it presents a much better control of the optics required to couple light to electrons and record the signals, generating ~ 100 microeV resolution spectra (FWHM). The experiments are very innovative, in the sense of the instrumentation (compared to previous EEG spectroscopy), and the demonstration of the resolution on WGM resonators.

While I am very impressed by the resolution, signals, and approach in absolute terms, I am somewhat concerned that the title and introduction convey the impression that the technique is generally applicable as one could do spectromicroscopy as EELS Spectrum imaging but this applies specifically to cases where the coupling between light and photons spectroscopy, can be maximized. How much would this approach work on really any samples? Can one do spectrum imaging on a sample to study local defects? What are the necessary requirements to get this kind of resolution for "any samples"?

In the first paragraphs, the authors state:

" we demonstrate high-spectral-resolution EEGS enhanced by mode matching between a free-space laser beam and the sample, which renders the technique generally applicable to any kind of specimen."

However, is this really generally applicable? WGM are specific case where EEG signal coupling is enhanced by mode matching (and high Q factor) by several orders of magnitude (even stated in the abstract). How does this work for a general sample? Is the second part of that sentence ("... which renders....") really demonstrated? I do not think this is done and suggest that the authors demonstrate this or remove that part of the sentence. The title should also be more specific unless the generality is demonstrated.

Answer to the reviewers:

Reviewer #1 (Remarks to the Author):

The manuscript by Auad et al. reports fascinating experiments on electron-energy gain spectroscopy (EEGS) at a resonant dielectric nanostructure. In the experiments, a monochromated electron beam from a designated scanning transmission electron microscope (STEM) is passed by optically illuminated dielectric spheres. These structures facilitate the inelastic interaction of the electrons with excited whispering gallery modes, which results in stimulated scattering and the generation of single sidebands in the electron energy spectrum (as in so-called PINEM experiments).

The present study varies the frequency of the incident nanosecond laser across different resonances in the structure and records the fraction of electrons inelastically scattered, which promotes a higher resolution than that of the microscope by a form of “excitation spectroscopy”, which was previously coined EEGS (by two of the authors, in ref. [14]).

The technical development, measured data, and overall manuscript presentation are of excellent quality and, in my opinion, clearly merit publication in Nature Communications. A particularly interesting physical aspect concerns the mode-matched excitation of the dielectric resonators by focusing the laser in close vicinity of the surface. I believe the work will be of considerable interest to readers from both the immediate community and further subject areas.

The authors thank reviewer for his comment.

I suggest the authors address the following minor comments in a revision:

1) The authors state that the approach allows them to “interrogate any type of photonic structure”, and that it is “generally applicable to any kind of specimen” and “arbitrary structures“. I fully agree with the authors on the impressive progress documented by this work, but these specific phrases might be slight overstatements. I believe it is not a coincidence that the present work considers structures that resemble characteristic features of Refs. [13] and [20]. Specifically, the combination of phase matching and resonant enhancement does seem to be practical or possibly necessary for the observability of the phenomenon with present technical capabilities. Possibly, the authors could slightly balance these formulations and also link their results a bit better to these recent observations.

We agree with the comments of the reviewer. In fact, we meant that the technical/instrumental aspects of the experiment are generally applicable to any *photonic* (including plasmonic) system. In particular, the same set up can be used to study low to very high Q system. This versatility was part of what we

implied for “arbitrary”, and we have now made it clearer in the text. But more importantly, we were considering high Q cavities. Indeed, the true fact that Q is high means that radiative coupling to the far field is weak and in the general case, coupling to these modes with light requires near field strategies. Mode matching singularly breaks this dependence, meaning it allows to study these systems without having to develop specific samples. We already pointed this in the former version of the paper, but missed to mention it in a critical section. We have therefore made the following modification in the paragraph at the end of the introduction:

“Here, we demonstrate high-spectral-resolution EEGS **using free-space light injection** enhanced by mode matching between a free-space laser beam and the sample, which renders the technique generally applicable to ~~any kind of specimen~~ **a broad variety of photonic specimens.**”

2) Panel 1c shows data points separated by 7 μeV . It is not fully clear from the figure and caption that this panel is also a schematic and not a measurement, and that the 7 μeV is the linewidth of the laser.

This was clearly misleading. This information is now clearer in the paper with a modified caption for figure 1.

“(B) A series of EELS spectra is acquired for a given electron probe position as the laser wavelength is scanned over the spectral region of interest (**simulated data**). The strength of the energy-gain signal is boosted when the light wavelength is close to an optical resonance of the specimen. (C) ~~Using a wavelength-independent light power,~~ **Schematic of a resonance probed by EEGS in a constant power regime.** The energy-gain signal yields the EEGS intensity as a function of light wavelength, **only limited by the nominal laser linewidth (7 μeV for the laser used in this study).**”

3) The abstract states that the authors carried out a “nanoscale mapping of photonic modes”, but the manuscript does not show any spatial scans of the electron beam. Have the authors also carried out such measurements? This should probably be addressed. Can they comment on opportunities and challenges towards EEGS-imaging?

Indeed, this phrase looks like an overstatement. We have changed to:

“Here, we demonstrate $\mu\text{electronvolt}$ spectral resolution ~~in the nanoscale mapping~~ **with a sub-nanometer probe** of photonic modes with quality factors as high as 10^4 .”

Spatially-resolved measurements were tried in such structures, but charging effects undermines their reproducibility.

Reviewer #2 (Remarks to the Author):

Manuscript by Y. Auad et al, "microeV electron"

This manuscript presents a novel implementation of the energy gain spectroscopy approach, improving the energy resolution and signals by order of magnitudes compared to previous work. The work is exciting as it presents a much better control of the optics required to couple light to electrons and record the signals, generating ~ 100 microeV resolution spectra (FWHM). The experiments are very innovative, in the sense of the instrumentation (compared to previous EEG spectroscopy), and the demonstration of the resolution on WGM resonators.

The authors thank reviewer for their comment.

While I am very impressed by the resolution, signals, and approach in absolute terms, I am somewhat concerned that the title and introduction convey the impression that the technique is generally applicable as one could do spectromicroscopy as EELS Spectrum imaging but this applies specifically to cases where the coupling between light and photons spectroscopy, can be maximized. How much would this approach work on really any samples? Can one do spectrum imaging on a sample to study local defects? What are the necessary requirements to get this kind of resolution for "any samples"?

This question is clearly in line with reviewer 1's comment, that we addressed above. In addition to this, the reviewer quite relevantly points out the applicability of the technique to "points defects". We can extend the question to what is sometimes referred as "incoherent excitations" (JGA, Review of Modern physics 2010), which are the collection of excitations that are not collective in nature, such as excitons or electron-holes. For these systems, the induced electrical field in eq. 1 is much smaller than for collective excitations and it is unclear if the experiment is feasible. This question is also relevant to other related spectroscopies (such as PINEM) and has not been addressed so far to the best of our knowledge. We also believe this is an important point. Therefore, we have made a comment in the discussion on this specific point.

"Beyond, this also permits to envision the study of optical materials such as quantum-confined systems, 2D crystals or point defects. Nevertheless, we note that even in PINEM, no experiments have been reported on optical excitations in atomic-scale systems. This might be related to the low electrical field in equation 1 expected for excitons or electron-hole pairs compared to collective excitations such as plasmons of photonic modes. Extreme mode and energy-matching such as proposed here might be a key for the success of such investigation."

In the first paragraphs, the authors state:

“ we demonstrate high-spectral-resolution EEGS enhanced by mode matching between a free-space laser beam and the sample, which renders the technique generally applicable to any kind of specimen.”

However, is this really generally applicable? WGM are specific case where EEG signal coupling is enhanced by mode matching (and high Q factor) by several orders of magnitude (even stated in the abstract). How does this work for a general sample? Is the second part of that sentence (“... which renders....”) really demonstrated? I do not think this is done and suggest that the authors demonstrate this or remove that part of the sentence. The title should also be more specific unless the generality is demonstrated.

As discussed in the previous answer, EEGS in high-Q cavities is in fact very difficult because light coupling is very inefficient. When we have stated that this setup can be applicable for an arbitrary sample, we meant that the technique itself can be used, and the fact that even with inefficient light coupling, EEGS signal can be retrieved by applying mode-matching or slight monochromation of the electron beam. We have commented a bit more on this absence of efficiency being one of the root of the absence of former experiments in free space. In passing, we also have made sure now that there is no confusion between mode matching (between the light beam and the WGM) and phase matching (between the electron and the WGM). Nevertheless, the authors agree with the reviewer that the phrase seemed an overstatement. We have rephrased the mentioned paragraph to:

“Here, we demonstrate high-spectral-resolution EEGS using free-space light injection, which renders the technique applicable to a broad variety of specimens.”

As for the title, we respectfully don't agree with the reviewer point of view. What we have been demonstrating is literally a spectromicroscopy that use free light for reaching μeV resolution. We already know it can be applied to high Q photonic structure; similar non-spectrally resolved experiments can be done on plasmons where the spectral resolution and coupling to light is the least of the concern, so the applicability to the field of photonics and plasmonic is already clear. But the main point is the possibility to reach μeV resolution in free space, which is clearly demonstrated it. We therefore advocate keeping the title as it is.

REVIEWERS' COMMENTS

Reviewer #1 (Remarks to the Author):

The authors carefully considered all questions raised by the Reviewers, and they revised the manuscript accordingly. I congratulate the authors to these results and enthusiastically recommend the manuscript for publication!

(Final note: I noticed from the revision adding phase matching that the manuscript never mentions that PINEM on such structures, facilitated by resonant enhancement and phase matching, was demonstrated in Ref. [13].)

Reviewer #2 (Remarks to the Author):

Thank you for considering the comments and making changes. I have no problems with the document. Thank you.

Answer to the reviewers:

There was only one comment from reviewer, mentioning we did not mention that in ref 13 the same objects were already used for PINEM (we mentioned for EELS and CL). This has been corrected:

To illustrate and validate the principle of EEGS and its relation to other spectroscopies, we start by studying $\sim 4 \mu\text{m}$ silica spheres (Figure 2), which are known to exhibit under similar experimental conditions quality factors $Q \sim 102$ and good electron-WGM coupling for the employed 200 keV electrons [21], and have been successfully studied by PINEM [13]